

# Bioinformatic prediction of immunodominant regions in spike protein for early diagnosis of the severe acute respiratory syndrome coronavirus 2 (SARS-CoV-2)

Siqi Zhuang[1], Lingli Tang[1], Yufeng Dai[1], Xiaojing Feng[1], Yiyuan Fang[1], Haoneng Tang[1], Ping Jiang[1], Xiang Wu[2], Hezhi Fang[3] and Hongzhi Chen[4]

[1] Department of Laboratory Medicine, The Second Xiangya Hospital, Central South University, Changsha, Hunan, China
[2] Department of Parasitology, Xiangya School of Basic Medicine, Central South University, Changsha, Hunan, China
[3] Key Laboratory of Laboratory Medicine, Ministry of Education, Zhejiang Provincial Key Laboratory of Medical Genetics, College of Laboratory Medicine and Life Sciences, Wenzhou Medical University, Wenzhou, Zhejiang, China
[4] National Clinical Research Center for Metabolic Disease, Key Laboratory of Diabetes Immunology, Ministry of Education, Metabolic Syndrome Research Center, and Department of Metabolism & Endocrinology, The Second Xiangya Hospital, Central South University, Changsha, Hunan, China

Corresponding author
Hongzhi Chen,
chenhongzhi2013@csu.edu.cn

## ABSTRACT

**Background:** To contain the pandemics caused by SARS-CoV-2, early detection approaches with high accuracy and accessibility are critical. Generating an antigen-capture based detection system would be an ideal strategy complementing the current methods based on nucleic acids and antibody detection. The spike protein is found on the outside of virus particles and appropriate for antigen detection.

**Methods:** In this study, we utilized bioinformatics approaches to explore the immunodominant fragments on spike protein of SARS-CoV-2.

**Results:** The S1 subunit of spike protein was identified with higher sequence specificity. Three immunodominant fragments, Spike$_{56-94}$, Spike$_{199-264}$, and Spike$_{577-612}$, located at the S1 subunit were finally selected via bioinformatics analysis. The glycosylation sites and high-frequency mutation sites on spike protein were circumvented in the antigen design. All the identified fragments present qualified antigenicity, hydrophilicity, and surface accessibility. A recombinant antigen with a length of 194 amino acids (aa) consisting of the selected immunodominant fragments as well as a universal Th epitope was finally constructed.

**Conclusion:** The recombinant peptide encoded by the construct contains multiple immunodominant epitopes, which is expected to stimulate a strong immune response in mice and generate qualified antibodies for SARS-CoV-2 detection.

## INTRODUCTION

The severe acute respiratory syndrome coronavirus 2 (SARS-CoV-2) is highly contagious and has caused more than one hundred million infection cases and over 2.4 million deaths (https://www.who.int/, as of February 15, 2021), posing a huge economic and social burden internationally (*Lan et al., 2020*; *Shang et al., 2020*). The reports of SARS-CoV-2 reinfection cases suggest that stronger international efforts are required to prevent COVID-19 re-emergence in the future (*Zhan, Deverman & Chan, 2020*). Nevertheless, the possibility of SARS-CoV-2 becoming a seasonal epidemic cannot be excluded (*Shaman & Galanti, 2020*). Even worse, the large number of asymptomatic infections greatly increase the difficulties of epidemic control (*Rothe et al., 2020*). At present, no specific drugs have been developed for SARS-CoV-2, and the effectiveness of the vaccines on the market still needs time to be evaluated. Therefore, early detection and isolation of infected people are still indispensable means to control the spread of the epidemic, which requires accurate, early, economical, and easy-to-operate diagnostic methods (*Yan, Chang & Wang, 2020*).

The real-time reverse transcriptase-polymerase chain reaction (RT-PCR) and antibody-capture serological tests are currently the main diagnostic methods for SARS-CoV-2 (*Ishige et al., 2020*). As the golden standard, RT-PCR is highly reliable (*Bustin & Nolan, 2020*; *Padoan et al., 2020*). However, the implementation costs and relatively cumbersome operation problems make it a big challenge for large population screening (*Thabet et al., 2020*). The antibody-capture serological test is convenient, but seroconversion generally occurs in the second or third week of illness. Therefore, it is not ideal for the early diagnosis of infection (*Hachim et al., 2020*; *Liu et al., 2020*; *Tang et al., 2020*). The antigen-capture test is an alternative diagnostic method that relies on the immunodetection of viral antigens in clinical samples. Accordingly, this method could be applied for the detection of early infection no matter if the patient was asymptomatic or not (*Ohnishi, 2008*). Compared with RT-PCR based detection method, it is relatively inexpensive and can be used at the point-of-care.

Rapid viral antigen detection has been successfully used for diagnosing respiratory viruses such as influenza and respiratory syncytial viruses (*Cazares et al., 2020*; *Ji et al., 2011*; *Ohnishi et al., 2005*, *2012*; *Qiu et al., 2005*). The sensitivity and specificity of the antigen-capture detection system depend highly on the antigen employed to generate antibodies (*Ohnishi et al., 2012*). The spike protein is one of the structural proteins of SARS-CoV-2, with the majority located on the outside surface of the viral particles (*Fehr & Perlman, 2015*; *Kumar et al., 2020*; *Woo et al., 2005*). It has a 76.4% homology with the spike protein of SARS-CoV. *Sunwoo et al. (2013)* showed that the bi-specific spike protein derived monoclonal antibody system exhibited excellent sensitivity in SARS-CoV detection. The virus infection is initiated by the interaction of spike protein receptor-binding domain (RBD) and angiotensin-converting enzyme 2 (ACE2) on host cells. It is

widely accepted that the spike protein is one of the earliest antigenic proteins recognized by the host immune system (*Callebaut, Enjuanes & Pensaert, 1996*; *Chen et al., 2020c*; *Gomez et al., 1998*; *Lu et al., 2004*; *Sanchez et al., 1999*). Nevertheless, the difficulties of using spike protein as an antigen are also obvious. Firstly, it is not easy to express and purify the full-length spike protein (*Tan et al., 2004*). Besides, the spike protein is highly glycosylated (*Kumar et al., 2020*) and prone to mutation (*Wang et al., 2020a*), which may counteract the sensitivity of antigen-capture based detection method. Hence, it is critical to truncating the glycosylation and mutation sites on spike protein as much as possible in antigen design (*Meyer, Drosten & Muller, 2014*; *Tan et al., 2004*). A study using the truncated spike protein to detect SARS-CoV achieved a diagnostic sensitivity of >99% and a specificity of 100% (*Mu et al., 2008*), which suggests that the truncated spike protein of SARS-CoV-2 could also be an appropriate candidate for the early diagnostic testing and screening of SARS-CoV-2. In this study, we analyzed the spike protein via bioinformatics tools to obtain immunodominant fragments. The predicted sequences were joined together as a novel antigen for the immunization of mice and antibody production. Epitopes information presented by this work may aid in developing a promising antigen-capture based detection system in pandemic surveillance and containment.

## METHOD

### Data retrieval and sequence alignment

Multiple bioinformatics analysis tools were used in this study, and the flowchart is depicted in Fig. 1. Coronaviruses had four genera composed of alpha-, beta-, gamma- and delta-coronaviruses. Among them, alpha- and beta- genera could infect humans. Seven beta-coronaviruses are known to infect humans (HCoV-229E, HCoV-OC43, HCoV-NL63, HCoV-HKU1, SARS-CoV, MERS-CoV, and SARS-CoV-2) (*Kin et al., 2015*; *Su et al., 2016*). We utilized the NCBI database to obtain the sequences of these human-related coronaviruses spike proteins, of which accession numbers were presented in Fig. 2A. The Clustal Omega Server-Multiple Sequence Alignment was used to analyze the sequence similarity. The analysis of the phylogenetic tree was calculated by the same server. In this study, we set parameters of Clustal Omega as default (*Sievers et al., 2011*). Additionally, we conducted the EMBOSS Needle Server-Pairwise Sequence Alignment (*Needleman & Wunsch, 1970*) to compare the whole sequence and several major domains between SARS-CoV-2 and SARS-CoV to find out the specific genomic regions on SARS-CoV-2.

### Linear B-cell epitope prediction

Linear B-cell epitopes of the SARS-CoV-2 spike protein were calculated by ABCpred and Bepipred v2.0 servers. For ABCpred, we set a threshold of 0.8 to achieve a specificity of 95.50% and an accuracy of 65.37% for prediction. The window length was set to 16 (the default window length) in this study (*Saha & Raghava, 2006*). The BepiPred v2.0 combines a hidden Markov model and a propensity scale method. The score threshold for the BepiPred v2.0 was set to 0.5 (the default value) to obtain a specificity of 57.16% and a
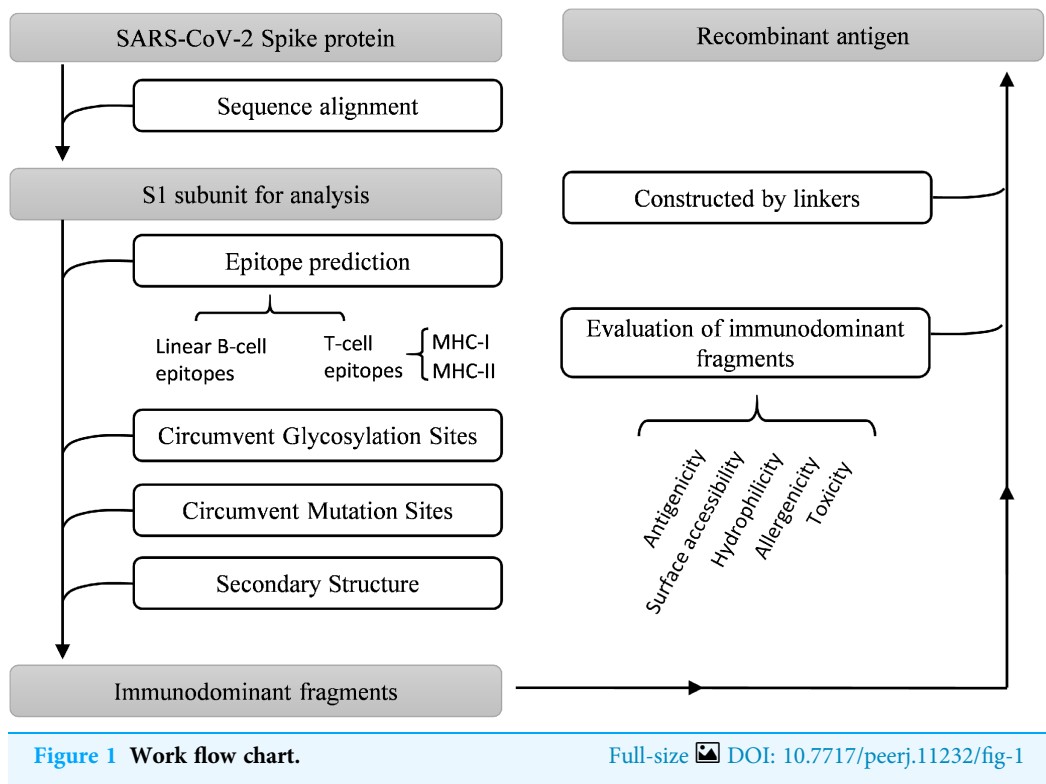

**Figure 1 Work flow chart.**

sensitivity of 58.56% (*Jespersen et al., 2017*). The residues with scores above 0.5 were predicted to be part of an epitope.

## T-cell epitope prediction

The free online service TepiTool server, integrated into the Immune Epitope Database (IEDB), was used to forecast epitopes binding to mice MHC molecules (*Paul et al., 2016*). Alleles including H-2-Db, H-2-Dd, H-2-Kb, H-2-Kd, H-2-Kk, and H-2-Ld were selected for MHC-I binding epitopes analysis. We checked the "IEDB recommended" option during computation and retained sequences with predicted consensus percentile rank ≤1 as predicted epitope (*Trolle et al., 2015*). For MHC-II binding epitopes, alleles including H2-IAb, H2-IAd, and H2-IEd were selected for analysis. As the same as MHC-I binding computation, we chose the "IEDB recommended" option, and peptides with predicted consensus percentile rank ≤10 were identified as potential epitopes (*Wang et al., 2010*; *Zhang et al., 2012*).

## Profiling and evaluation of selected fragments

The secondary structure of the SARS-CoV-2 spike protein (PDB ID: 6VSB chain B) was calculated by the PyMOL molecular graphics system using the SSP algorithm. PyMOL (http://www.pymol.org) is a python-based tool, which is widely used for visualization of macromolecules, such as SARS-CoV-2 spike protein in the current study (*Yuan et al., 2016*). Vaxijen2.0 server was utilized to analyze the antigenicity of epitopes and selected fragments. A default threshold of 0.4 was set and the prediction accuracy is between 70% and 89% (*Doytchinova & Flower, 2007*). The hydrophilicity of the selected fragment was
**A**

| Virus | Accession ID | Identity compared to SARS-CoV-2(%) |
|---|---|---|
| SARS-CoV-2 | QIC53213.1 | 100 |
| SARS-COV | ABD72968.1 | 77.46 |
| MERS-COV | QBM11748.1 | 32.04 |
| HCoV-NL63 | YP_003767.1 | 25.57 |
| HCoV-229E | AWH62679.1 | 26.90 |
| HCoV-OC43 | AIX10763.1 | 30.92 |
| HCoV-HKU1 | AMN88694.1 | 29.92 |

**B**

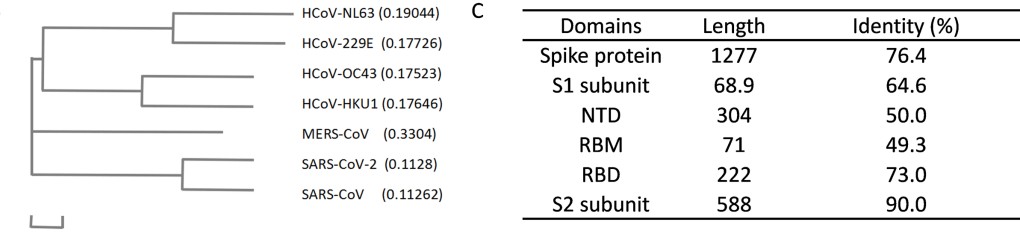

**C**

| Domains | Length | Identity (%) |
|---|---|---|
| Spike protein | 1277 | 76.4 |
| S1 subunit | 68.9 | 64.6 |
| NTD | 304 | 50.0 |
| RBM | 71 | 49.3 |
| RBD | 222 | 73.0 |
| S2 subunit | 588 | 90.0 |

**D**

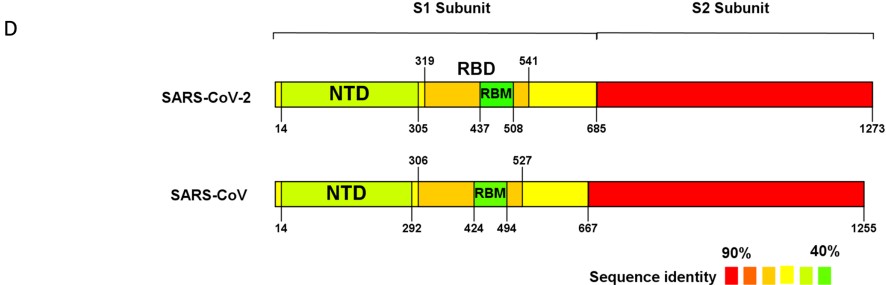

**Figure 2 Sequence alignment results of spike protein.** (A) Accession IDs and sequence identities of selected coronavirus spike protein. (B) Phylogenetic tree of spike proteins among selected coronavirus. (C) Sequence identity of major domains in spike protein between SARS-CoV-2 and SARS-CoV. (D) Sequence identity of domains in SARS-CoV-2 and SARS-CoV reflected by colors. From red to green, the color changing represents the sequence identity from high to low.

analyzed by the online server ProtScale (*Wilkins et al., 1999*). Surface accessibility of predicted fragments was evaluated by NetsurfP, an online server calculating the surface accessibility and secondary structure of amino acid sequence (*Petersen et al., 2009*). Critical features such as allergenicity and toxicity were evaluated by online server AllerTOP v2.0 (*Dimitrov et al., 2014*) and ToxinPred (*Gupta et al., 2013*). In addition, we utilized IEDB (www.iedb.org) to search the selected fragments and epitopes to clarify whether these peptides have been experimentally verified (*Vita et al., 2019*). Protein sequence BLAST was performed to evaluate the possibility of cross-reactivity with other mouse protein sequences (*Altschul et al., 1997*).

## RESULTS

### Sequence alignment of spike protein in different coronaviruses

We performed sequence alignment to determine the evolutionary relationships between SARS-CoV-2 and other beta-coronaviruses that could infect humans. According to the results of sequence alignment (Figs. 2A and 2B), SARS-CoV is the closest virus to SARS-CoV-2 among the seven HCoVs, exhibiting a 77.46% sequence identity. To better

understand the divergence of spike protein sequences between SARS-CoV-2 and SARS-CoV, we further analyzed the sequences of main domains. Results showed that the S2 subunit was the most conserved domain with a 90.0% identity. RBM and NTD domains, which were located in the S1 subunit, exhibited 49.3% and 50.0% identity respectively (Figs. 2C and 2D). Hence, we chose the S1 subunit (amino acid 1-685) for the subsequent bioinformatics analysis given their high specificity.

## Linear B-cell epitope prediction of S1 subunit in SARS-CoV-2 spike protein

The B-cell epitope is a surface accessible cluster of amino acids, which could be recognized by secreted antibodies or B-cell receptors and elicit humoral immune response (*Getzoff et al., 1988*). The immunodominant fragments should contain high-quality linear B-cell epitopes to stimulate antibody production effectively. The sequence of the SARS-CoV-2 S1 subunit was evaluated via ABCpred and BepiPred v2.0. A total of 31 peptides were identified by the ABCpred algorithm (Table S1). For the Bepipred v2.0 server, 14 epitopes were forecasted (Table S2). After antigenicity evaluation, 19 and 9 potential linear B-cell epitopes predicted by the ABCpred server and BepiPred v2.0 server were obtained respectively (Table 1). The peptides predicted by both bioinformatics programs are more likely to be an epitope recognized in vivo. After mapping the positions of peptides identified by these servers, 3 regions containing predicted epitopes were obtained. These regions could be preliminarily considered as candidates for immunodominant fragments (Fig. 3; Table 2).

## Murine T-cell epitope prediction of S1 subunit in SARS-CoV-2 spike protein

Though B cells are responsible for producing antibodies, humoral immunity is heavily dependent on the activation of T cells (*Cho et al., 2019a*). Helper T cells (Th) recognize antigen peptides presented by MHC-II molecules and facilitate the humoral immune response (*Cho et al., 2019b*; *Mahon et al., 1995*). During humoral immune responses, antigen-activated T cells could provide help in many aspects including directing antibody class switching and guiding the differentiation of antibody-secreting plasma cells as well as the properties of the B-cell antigen receptor (*Cho et al., 2019a*; *Paus et al., 2006*; *Shulman et al., 2014*). Therefore, the immunodominant fragments containing T-cell epitopes could offer essential help to powerful antibody production. The S1 subunit was selected for the prediction of T-cell epitopes. We utilized the TepiTool server to forecast MHC-I and MHC-II binding epitopes. A total of 35 MHC-I binding epitopes was predicted (Table S3), and 27 peptides were identified as MHC-II binding epitopes (Table S4). The antigenicity of these peptides was calculated via Vaxijen 2.0 server (Table 3). Combined with the MHC-II epitopes prediction results, the candidate immunodominant fragments were adjusted (Fig. 4). Compared with the preliminary candidate immunodominant fragments screened according to the linear B-cell epitope prediction, we added the Spike$_{14-34}$ fragment into consideration because it contains a linear

**Table 1 Linear B-cell epitopes predicted by ABCpred and BepiPred v2.0 with antigenicity score exceed the threshold value.**

| Tools | Position | Sequence | Length | Antigenicity (cut off ≥ 0.4) |
|---|---|---|---|---|
| ABCpred | 583-598 | EILDITPCSFGGVSVI | 16 | 1.3971 |
| | 406-421 | EVRQIAPGQTGKIADY | 16 | 1.3837 |
| | 415-430 | TGKIADYNYKLPDDFT | 16 | 0.9642 |
| | 648-663 | GCLIGAEHVNNSYECD | 16 | 0.848 |
| | 288-303 | AVDCALDPLSETKCTL | 16 | 0.7905 |
| | 604-619 | TSNQVAVLYQDVNCTE | 16 | 0.7593 |
| | 307-322 | TVEKGIYQTSNFRVQP | 16 | 0.6733 |
| | 200-215 | YFKIYSKHTPINLVRD | 16 | 0.657 |
| | 257-272 | GWTAGAAAYYVGYLQP | 16 | 0.621 |
| | 329-344 | FPNITNLCPFGEVFNA | 16 | 0.6058 |
| | 245-260 | HRSYLTPGDSSSGWTA | 16 | 0.6017 |
| | 280-295 | NENGTITDAVDCALDP | 16 | 0.5804 |
| | 49-64 | HSTQDLFLPFFSNVTW | 16 | 0.5305 |
| | 492-507 | LQSYGFQPTNGVGYQP | 16 | 0.5258 |
| | 70-85 | VSGTNGTKRFDNPVLP | 16 | 0.5162 |
| | 236-251 | TRFQTLLALHRSYLTP | 16 | 0.5115 |
| | 266-281 | YVGYLQPRTFLLKYNE | 16 | 0.5108 |
| | 594-609 | GVSVITPGTNTSNQVA | 16 | 0.4651 |
| | 320-335 | VQPTESIVRFPNITNL | 16 | 0.4454 |
| Bepipred v2.0 | 179-190 | LEGKQGNFKNLR | 12 | 1.1188 |
| | 404-426 | GDEVRQIAPGQTGKIADYNYKLP | 23 | 1.1017 |
| | 14-34 | QCVNLTTRTQLPPAYTNSFTR | 21 | 0.7594 |
| | 56-81 | LPFFSNVTWFHAIHVSGTNGTKRFDN | 26 | 0.6041 |
| | 208-222 | TPINLVRDLPQGFSA | 15 | 0.5531 |
| | 141-160 | LGVYYHKNNKSWMESEFRVY | 20 | 0.5308 |
| | 249-261 | LTPGDSSSGWTAG | 13 | 0.495 |
| | 306-321 | FTVEKGIYQTSNFRVQ | 16 | 0.4361 |
| | 615-644 | VNCTEVPVAIHADQLTPTWRVYSTGSNVFQ | 30 | 0.4259 |

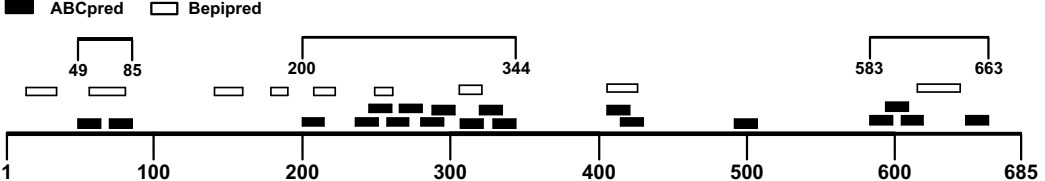

**Figure 3 Preliminary immunodominant fragments based on B-cell epitope prediction results.** The black squares represent epitopes predicted by ABCpred server, the black frames represent epitopes predicted by Bepipred v2.0 server, and the black lines with numbers on both ends represent the preliminary candidate immunodominant fragments.
**Table 2 Details of epitopes in the preliminary immunodominant fragments selected according to linear B-cell epitope prediction results.**

| Regions | Epitope predicted by ABCpred | | | Epitope predicted by Bepipred v2.0 | | |
|---|---|---|---|---|---|---|
| | Position | Sequence | Antigenicity | Position | Sequence | Antigenicity |
| 49-85 | 49-64 | HSTQDLFLPFFSNVTW | 0.5305 | 56-81 | LPFFSNVTWFHAIHVSGTNGTKRFDN | 0.6041 |
| | 70-85 | VSGTNGTKRFDNPVLP | 0.5162 | | | |
| 200-344 | 200-215 | YFKIYSKHTPINLVRD | 0.6570 | 208-222 | TPINLVRDLPQGFSA | 0.5531 |
| | 236-251 | TRFQTLLALHRSYLTP | 0.5115 | 249-261 | LTPGDSSSGWTAG | 0.4950 |
| | 245-260 | HRSYLTPGDSSSGWTA | 0.6017 | | | |
| | 257-272 | GWTAGAAAYYVGYLQP | 0.6210 | | | |
| | 266-281 | YVGYLQPRTFLLKYNE | 0.5108 | | | |
| | 280-295 | NENGTITDAVDCALDP | 0.5804 | | | |
| | 288-303 | AVDCALDPLSETKCTL | 0.7905 | 306-321 | FTVEKGIYQTSNFRVQ | 0.4361 |
| | 307-322 | TVEKGIYQTSNFRVQP | 0.6733 | | | |
| | 320-335 | VQPTESIVRFPNITNL | 0.4454 | | | |
| | 329-344 | FPNITNLCPFGEVFNA | 0.6058 | | | |
| | 415-430 | TGKIADYNYKLPDDFT | 0.9642 | | | |
| 583-663 | 583-598 | EILDITPCSFGGVSVI | 1.3971 | 615-644 | VNCTEVPVAIHADQLTPTWRVYSTGSNVFQ | 0.4259 |
| | 594-609 | GVSVITPGTNTSNQVA | 0.4651 | | | |
| | 604-619 | TSNQVAVLYQDVNCTE | 0.7593 | | | |
| | 648-663 | GCLIGAEHVNNSYECD | 0.8480 | | | |

B epitope and an MHC-II binding epitope, both of which had high antigenicity scores (Table 4).

## Immunodominant fragments refinement according to the glycosylation site distribution, mutation site distribution, and secondary structure

A profile of 24 glycosylation sites of SARS-CoV-2 spike protein has been reported (*Shajahan et al., 2020*). Since glycans could hinder the recognition of antigens by shielding the residues (*Walls et al., 2019*), protein glycosylation would affect the performance of antigen detection. Thus, glycosylation sites should be circumvented when selecting the immunodominant fragments. According to the study of *Shajahan et al. (2020)*, 15 glycosylation sites were located in the S1 subunit of the spike protein. Hence, the fragments in this study were adjusted to Spike$_{14-34}$, Spike$_{49-101}$, Spike$_{199-261}$, and Spike$_{583-620}$. To retain antigenicity of the epitopes, the final identified fragments only contained 3 glycosylation sites which should have a minimum effect on antigen recognition.

Rapid transmission of COVID-19 provides the SARS-CoV-2 with substantial opportunities for natural selection and mutations. To ensure the stability of the detection method, the immunodominant fragments were modified to avoid high-frequency mutation sites (*Wang et al., 2020b*). Spike$_{14-34}$ were excluded for containing four high-frequency mutation sites. Fragment Spike$_{49-101}$ was adjusted to Spike$_{56-92}$, and fragment Spike$_{583-620}$ was adjusted to Spike$_{583-609}$. By adjusting the fragments, we avoided

**Table 3 MHC-II and MHC-I binding epitopes predicted by TepiTool server with antigenicity score exceed threshold value.**

| Type | Position | Sequence | Length | Allele | Core (smm-align) | Core (nn-align) | Percentile Rank | Antigenicity (cut off ≥ 0.4 |
|---|---|---|---|---|---|---|---|---|
| MHC-II binding | 538-552 | CVNFNFNGLTGTGVL | 15 | H2-IAb | FNFNGLTGT | FNFNGLTGT | 8.55 | 1.3281 |
| | 374-388 | FSTFKCYGVSPTKLN | 15 | H2-IAb | FKCYGVSPT | YGVSPTKLN | 6.45 | 1.0042 |
| | 199-213 | GYFKIYSKHTPINLV | 15 | H2-Iab | KIYSKHTPI | YSKHTPINL | 6.9 | 0.9278 |
| | 18-32 | LTTRTQLPPAYTNSF | 15 | H2-IAb | TRTQLPPAY | TRTQLPPAY | 9.9 | 0.79 |
| | 60-74 | SNVTWFHAIHVSGTN | 15 | H2-IAb | VTWFHAIHV | TWFHAIHVS | 9.1 | 0.7044 |
| | 263-277 | AAYYVGYLQPRTFLL | 15 | H2-IAb | VGYLQPRTF | VGYLQPRTF | 8.75 | 0.6073 |
| | 592-606 | FGGVSVITPGTNTSN | 15 | H2-IAb | VITPGTNTS | VSVITPGTN | 6 | 0.5825 |
| | 238-252 | FQTLLALHRSYLTPG | 15 | H2-IEd | TLLALHRSY | TLLALHRSY | 9.85 | 0.5789 |
| | 345-359 | TRFASVYAWNRKRIS | 15 | H2-IAb | FASVYAWNR | YAWNRKRIS | 7.45 | 0.4963 |
| | 215-229 | DLPQGFSALEPLVDL | 15 | H2-IAb | FSALEPLVD | FSALEPLVD | 6.05 | 0.4812 |
| | 140-154 | FLGVYYHKNNKSWME | 15 | H2-IEd | GVYYHKNNK | YYHKNNKSW | 6.4 | 0.4793 |
| | 512-526 | VLSFELLHAPATVCG | 15 | H2-IAb | FELLHAPAT | FELLHAPAT | 2.9 | 0.4784 |
| | 87-101 | NDGVYFASTEKSNII | 15 | H2-Iab | YFASTEKSN | VYFASTEKS | 6.85 | 0.4277 |
| | 52-66 | QDLFLPFFSNVTWFH | 15 | H2-IAb | FLPFFSNVT | FLPFFSNVT | 2.95 | 0.4159 |
| | 233-247 | INITRFQTLLALHRS | 15 | H2-IAd | ITRFQTLLA | ITRFQTLLA | 1.9 | 0.4118 |
| MHC-I binding | 643-651 | FQTRAGCLI | 9 | H-2-Kk | | | 0.6 | 1.7332 |
| | 612-620 | YQDVNCTEV | 9 | H-2-Db | | | 0.4 | 1.6172 |
| | 539-547 | VNFNFNGLT | 9 | H-2-Kb | | | 0.47 | 1.5069 |
| | 503-511 | VGYQPYRVV | 9 | H-2-Kb | | | 0.47 | 1.4383 |
| | 379-387 | CYGVSPTKL | 9 | H-2-Kd | | | 0.3 | 1.4263 |
| | 16-24 | VNLTTRTQL | 9 | H-2-Kb | | | 0.86 | 1.3468 |
| | 510-518 | VVVLSFELL | 9 | H-2-Kb | | | 0.43 | 1.0909 |
| | 202-210 | KIYSKHTPI | 9 | H-2-Kb | | | 0.27 | 0.7455 |
| | 168-176 | FEYVSQPFL | 9 | H-2-Kk | | | 0.5 | 0.6324 |
| | 268-276 | GYLQPRTFL | 9 | H-2-Kd | | | 0.2 | 0.6082 |
| | 505-513 | YQPYRVVVL | 9 | H-2-Dd | | | 0.3 | 0.5964 |
| | 488-496 | CYFPLQSYG | 9 | H-2-Kd | | | 0.64 | 0.578 |
| | 215-223 | DLPQGFSAL | 9 | H-2-Dd | | | 0.69 | 0.5622 |
| | 342-350 | FNATRFASV | 9 | H-2-Kb | | | 0.56 | 0.5609 |
| | 84-92 | LPFNDGVYF | 9 | H-2-Ld | | | 0.21 | 0.5593 |
| | 484-492 | EGFNCYFPL | 9 | H-2-Kb | | | 0.84 | 0.5453 |
| | 62-70 | VTWFHAIHV | 9 | H-2-Kb | | | 0.61 | 0.5426 |
| | 489-497 | YFPLQSYGF | 9 | H-2-Dd | | | 0.8 | 0.5107 |
| | 350-358 | VYAWNRKRI | 9 | H-2-Kd | | | 0.7 | 0.5003 |
| | 60-68 | SNVTWFHAI | 9 | H-2-Kb | | | 0.82 | 0.4892 |
| | 262-270 | AAAYYVGYL | 9 | H-2-Kb | | | 0.98 | 0.4605 |

in a total of 8 high-frequency mutation sites (L5F, L18F, T29I, R21K/T, H49Y, L54F, S98F, D614G). The mainly mutant sites on the recent emergent highly infectious variants (including B.1.1.7, B.1.351, and P.1), such as N501Y, D614G, E484K, Y144del, K417N, and A570D were also not included in our fragments. The adjusted fragments contain none of

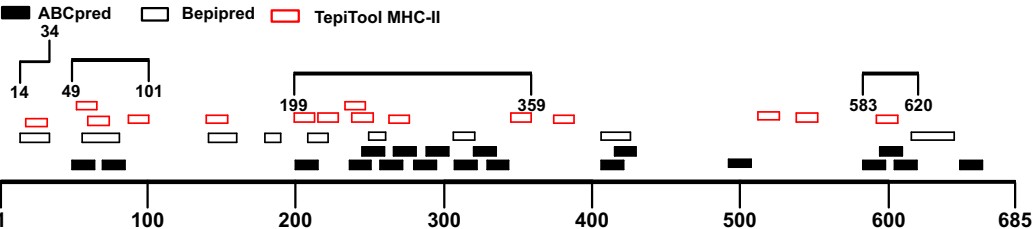

**Figure 4 Adjusted candidate immunodominant fragments according to MHC-II T-cell epitope prediction results.** The black squares represent epitopes predicted by ABCpred server, and the black frames represent epitopes predicted by Bepipred v2.0 server. The red frames denote MHC-II binding epitopes. The black lines with numbers on both ends represent the adjusted candidate fragments.

**Table 4 Details of candidate immunodominant fragments adjusted according to the MHC-II binding T-cell epitopes prediction results.**

| Regions | Linear B-cell epitopes | | | | MHC-II binding epitopes | | |
|---|---|---|---|---|---|---|---|
| | Tools | Position | Sequence | Antigenicity | Position | Sequence | Antigenicity |
| 14–34 | Bepipred v2.0 | 14–34 | QCVNLTTRTQLPPAYTNSFTR | 0.7594 | 18–32 | LTTRTQLPPAYTNSF | 0.7900 |
| 49–101 | Bepipred v2.0 | 56–81 | LPFFSNVTWFHAIHVSGTNGTKRFDN | 0.6041 | 52–66 | QDLFLPFFSNVTWFH | 0.4159 |
| | ABCpred | 49–64 | HSTQDLFLPFFSNVTW | 0.5305 | 60–74 | SNVTWFHAIHVSGTN | 0.7044 |
| | ABCpred | 70–85 | VSGTNGTKRFDNPVLP | 0.5162 | 87–101 | NDGVYFASTEKSNII | 0.4277 |
| 199–359 | Bepipred v2.0 | 208–222 | TPINLVRDLPQGFSA | 0.5531 | 199–213 | GYFKIYSKHTPINLV | 0.9278 |
| | | 249–261 | LTPGDSSSGWTAG | 0.4950 | | | |
| | | 306–321 | FTVEKGIYQTSNFRVQ | 0.4361 | | | |
| | ABCpred | 200–215 | YFKIYSKHTPINLVRD | 0.6570 | | | |
| | | 236–251 | TRFQTLLALHRSYLTP | 0.5115 | 215–229 | DLPQGFSALEPLVDL | 0.4812 |
| | | 245–260 | HRSYLTPGDSSSGWTA | 0.6017 | | | |
| | | 257–272 | GWTAGAAAYYVGYLQP | 0.6210 | 233–247 | INITRFQTLLALHRS | 0.4118 |
| | | 266–281 | YVGYLQPRTFLLKYNE | 0.5108 | | | |
| | | 280–295 | NENGTITDAVDCALDP | 0.5804 | 238–252 | FQTLLALHRSYLTPG | 0.5789 |
| | | 288–303 | AVDCALDPLSETKCTL | 0.7905 | | | |
| | | 307–322 | TVEKGIYQTSNFRVQP | 0.6733 | 263–277 | AAYYVGYLQPRTFLL | 0.6073 |
| | | 320–335 | VQPTESIVRFPNITNL | 0.4454 | | | |
| | | 329–344 | FPNITNLCPFGEVFNA | 0.6058 | 345–359 | TRFASVYAWNRKRIS | 0.4963 |
| 583–620 | ABCpred | 583–598 | EILDITPCSFGGVSVI | 1.3971 | 592–606 | FGGVSVITPGTNTSN | 0.5825 |
| | | 594–609 | GVSVITPGTNTSNQVA | 0.4651 | | | |
| | | 604–619 | TSNQVAVLYQDVNCTE | 0.7593 | | | |

the above high-frequency mutation sites, which might avoid the impact of mutations on detection performance and improve the detection efficiency in the future (*Li et al., 2020*; *Tegally et al., 2020*).

The PyMOL was used to present the secondary structure of the spike protein (PDB ID: 6VSB) (Fig. S1). To keep the integrity of the secondary structure of the selected fragments, we extended the N- and C-ends with 2~5 residues, and the immunodominant fragments were finally adjusted to Spike$_{56-94}$, Spike$_{199-264}$, and Spike$_{577-612}$. The epitopes

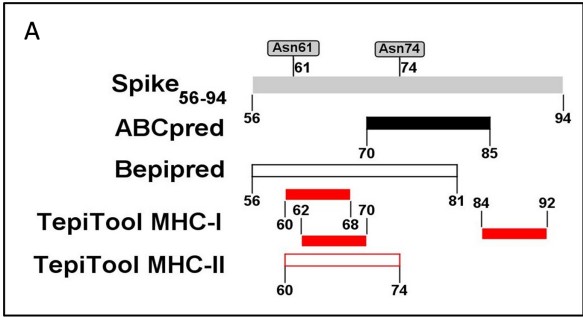

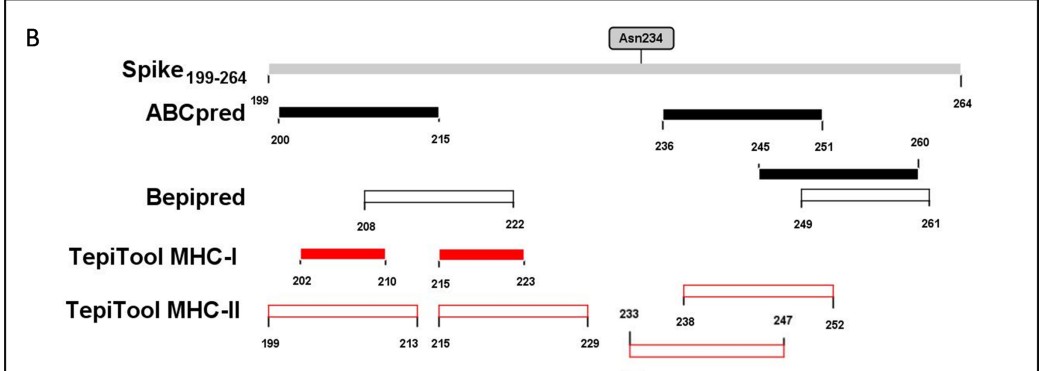

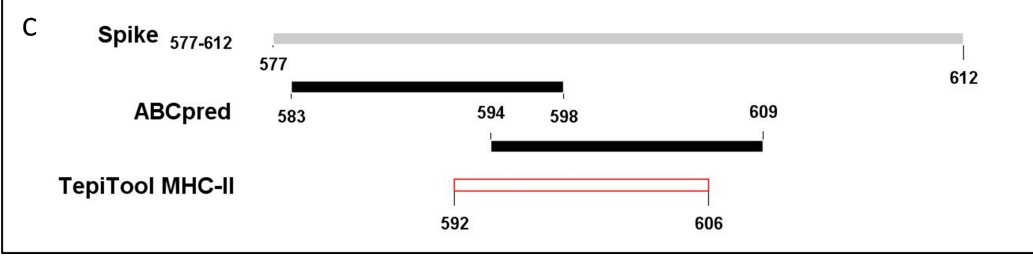

**Figure 5 The epitopes and glycosylation sites on the selected immunodominant fragments.** (A–C) Present the predicted epitopes and glycosylation sites on fragment Spike56-94, Spike199-264 and Spike577-612 respectively. The black squares represent epitopes predicted by ABCpred server, and the black frames represent epitopes predicted by Bepipred v2.0 server. The red squares represent MHC-I binding epitopes, and the red frames represent MHC-II binding epitopes. The gray squares mean occupied glycosylation sites contained in the selected fragments.

and potential glycosylation sites contained in the selected immunodominant fragments were displayed in Fig. 5.

## Profiling, evaluation, and visualization of selected immunodominant fragments

To further evaluate the antibody binding potentiality of these antigenic regions, the key features of the selected fragments such as antigenicity, hydrophilicity, surface accessibility, toxicity, and allergenicity were analyzed and presented (Table 5). The hydrophilicity and surface accessibility of the spike protein subunit 1 were calculated. The selected fragments of interest were submitted for computation of antigenicity, toxicity, and allergenicity. Three fragments presented relatively moderate hydrophilicity and surface

**Table 5 Significant features of the selected immunodominant fragments.** The sequences marked as bold and italic in the table represent amino acids with hydrophilicity and surface accessibility respectively.

| Fragments | Spike$_{56-94}$ | Spike$_{199-264}$ | Spike$_{577-612}$ |
|---|---|---|---|
| Length(aa) | 39 | 66 | 36 |
| Sequence | LPFFSNVTWFHAIHVSGTNGTKR FDNPVLPFNDGVYFAS | GYFKIYSKHTPINLVRDLPQGFSAL EPLVDLPIGINITRFQTLLALHRSYL TPGDSSSGWTAGAAA | RDPQTLEILDITPCSFGGV SVITPGTNTSNQVAVLY |
| Antigenicity | 0.4590 | 0.5774 | 0.9127 |
| Domain | S1(NTD) | S1(NTD) | S1 |
| Hydrophilicity fragments | LPFFSNVTWFHAIH*VSGTNGTKR FDNPVLP*FNDGV*YFAS* | *GYFKIYSKHTPIN*LV*RD*LPQGF *S*ALEPLVDLPIGINIT*RF*QTLL ALHRS*YLTPGDSSSGWT*AGAAA | *RDPQTL*EILDITPCSFGGVSVIT PG*TNTSN*QVA*V*LY |
| Surface Accessibility fragments | LP*FFSN*VTWFHAI*HV*S*GTNGTKR*F *D*NPVLP*FND*GVYFAS | GYFKIYSK *HTPINLVRD*LPQGF SAL E*PLV*D*LPIGIN*ITR*FQTLLALHRS *YLTPGDSSS*GWTAGAAA | R*DPQTL*EIL*DIT*PCS*FG*GVSVIT *PGTNTSNQ*VAVLY |
| Toxicity | Non-toxin | Non-toxin | Non-toxin |
| Allergenicity | non-allergen | non-allergen | probable allergen |

accessibility. The proportion of hydrophilic amino acids in the selected fragments Spike$_{56-94}$, Spike$_{199-264}$, Spike$_{577-612}$ are 48.72%, 45.45%, 33.33% respectively. The surface accessibility of these fragments calculated by the online server was shown in Table 5.

The toxicity of the selected fragments was examined and no fragment was predicted to be toxic. The allergenicity was assessed and only fragment Spike$_{577-612}$ was predicted to be a probable allergen. Attention should be paid to monitor potential allergic reactions when injecting the recombinant protein into mice. And the selected fragments were presented as the sphere in the trimer structure (Fig. 6). Next, we scanned the selected fragments utilizing the IEDB database to determine whether they were experimentally tested. The results showed that Spike$_{200-215}$(IEDB ID: 1330367) and Spike$_{238-252}$ (IEDB ID: 1329417) were identified experimentally as HLA class II epitope in SARS-CoV-2. Spike84-92 (IEDB ID: 1321049) and Spike202-210 (IEDB ID: 1319559) have been experimentally proved as HLA-B epitopes (Table S5). These findings enhanced the credibility of the current in silico analysis. The fragments identified would have a strong capacity in stimulating powerful antibody production.

## Immunodominant fragments based recombinant antigen design

Three immunodominant fragments embody several linear B-cell epitopes, MHC-I binding, and MHC-II binding T-cell epitopes were selected. As a universal Th epitope, the PAN DR epitope [PADRE(AKFVAAWTLKAAA)] was added into the construction aiming to boost helper T cell activity (*Alexander et al., 2000*; *Ghaffari-Nazari et al., 2015*). (GGGGS)$_n$ is a wildly used flexible linker with the function of segmenting protein fragments, maintaining protein conformation, preserving biological activity, and promoting protein expression (*Chen, Zaro & Shen, 2013*). Finally, we combined the fragments and a PADRE epitope by linker peptide (GGGGS)$_2$ and (GGGGS)$_3$ (*Chen, Zaro & Shen, 2013*) (Fig. 7). The predicted antigenicity of the final construct (194 aa)

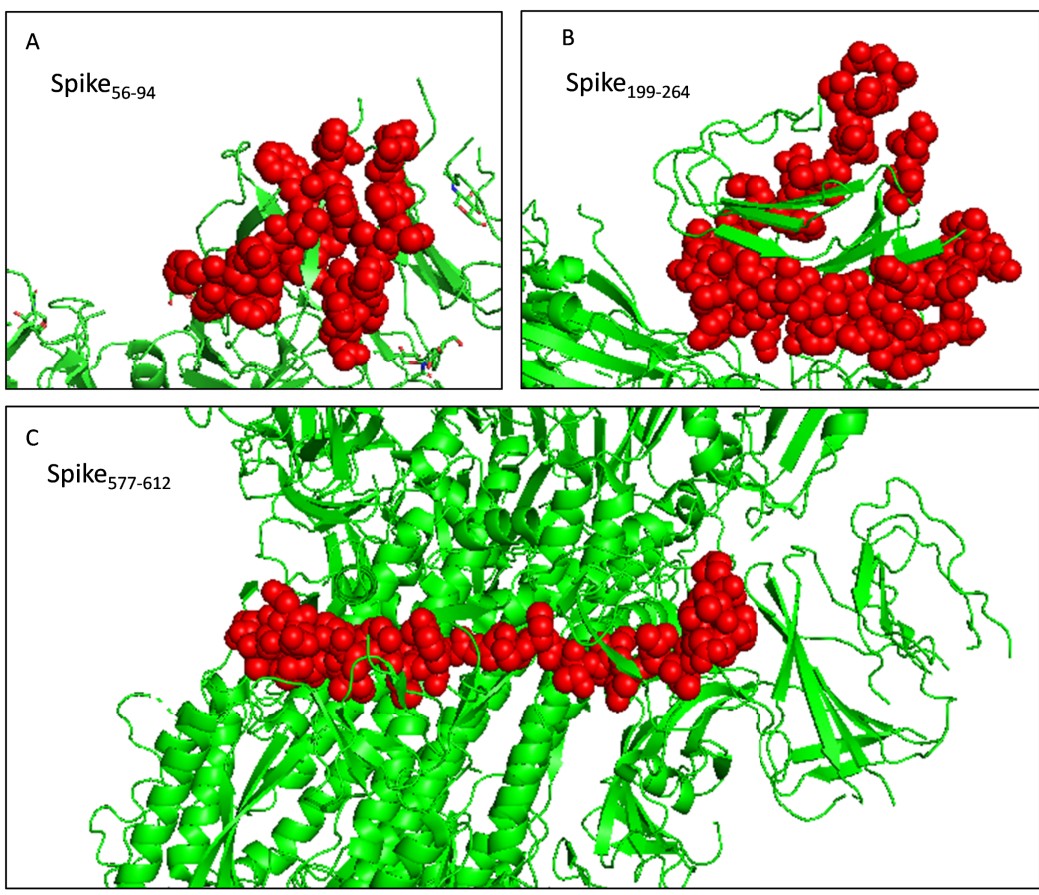

**Figure 6 Selected immunodominant fragments presented as spheres in the trimer structure of spike protein viewed by PyMOL.** Selected fragments were presented as red spheres, green cartoons denote unselected sections. (A, B, and C) denote fragments Spike$_{56-94}$, Spike$_{199-264}$, and Spike$_{577-612}$ respectively.

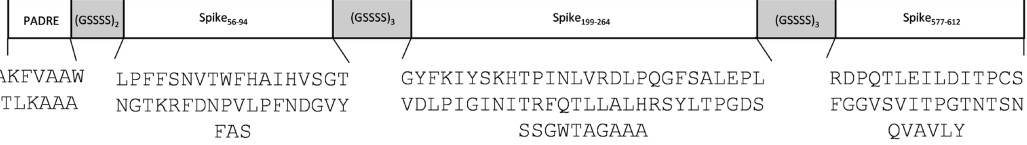

| PADRE | (GSSSS)$_2$ | Spike$_{56-94}$ | (GSSSS)$_3$ | Spike$_{199-264}$ | (GSSSS)$_3$ | Spike$_{577-612}$ |
|---|---|---|---|---|---|---|

```
AKFVAAW        LPFFSNVTWFHAIHVSGT        GYFKIYSKHTPINLVRDLPQGFSALEPL        RDPQTLEILDITPCS
TLKAAA         NGTKRFDNPVLPFNDGVY        VDLPIGINITRFQTLLALHRSYLTPGDS        FGGVSVITPGTNTSN
                      FAS                        SSGWTAGAAA                        QVAVLY
```

**Figure 7 A schematic diagram of recombinant peptide composed of selected fragments and a PADRE epitope.**

was 0.5690 (Table 6). A protein BLAST for the final construct was conducted to evaluate the possibility of cross-reactivity. The BLAST result suggested that, except for the SARS-CoV-2 spike protein, no protein would cross-react with the construct (Raw data in the Supplemental Files), which indicated that our fragments possess good specificity.

## DISCUSSION

In this study, the immunodominant fragments within the S1 subunit of the SARS-CoV-2 spike protein were explored. The final construct consists of three immunodominant fragments Spike$_{56-94}$, Spike$_{199-264}$, Spike$_{577-612}$, and a PADRE epitope. The recombinant

**Table 6 The structure and antigenicity of final recombinant peptides.**

| Final construct | PAN DR + (GGGGS)$_2$ + Spike$_{56-94}$ +(GGGGS)$_3$ + Spike$_{199-264}$ +(GGGGS)$_3$ + Spike$_{577-612}$ |
|---|---|
| Sequence | AKFVAAWTLKAAAGGGGSGGGGSLPFFSNVTWFHAIHVSGTNGTKRFDNPVLPFNDGVYFASGGGGSGGGGSGGGGSGYFKIYSKHTPINLVRDLPQGF SALEPLVDLPIGINITRFQTLLALHRSYLTPGDSSSGWTAGAAAGGGGSGGGGSGGGGSRDPQTLEILDITPCSFGGVSVITPGTNTSNQVAVLY |
| Antigenicity | 0.5690 |

antigen will be used to immunize mice to generate qualified antibody which could be applied for developing an antigen-capture based detection system.

The antibody-based antigen capturing method is user-friendly, time-saving, and economical. Thus, it is an ideal complementary detection strategy especially for early diagnosis and large population screening. The monoclonal antibodies against SARS-CoV have been successfully applied in the immunological antigen-detection of SARS-CoV (*Ohnishi, 2008*). Accordingly, we explored the immunodominant fragments on the spike protein of SARS-CoV-2, which would provide aid in developing an accurate and fast antigen-capture based early detection system for SARS-CoV-2.

We selected the S1 subunit for immunodominant fragments screening after divergence analysis. It had been reported that an S1 antigen-based assay of SARS-CoV could capture the virus as soon as the infection occurs (*Sunwoo et al., 2013*). Jong-Hwan Lee et al. designed a method that could seize and detect spike protein S1 subunit of SARS-CoV-2 using ACE2 receptor and S1-mAb (*Lee et al., 2021*). This finding suggests that it is appropriate to use the S1 subunit for specific and early diagnosis of SARS-CoV-2. Three immunodominant fragments (Spike$_{56-94}$, Spike$_{199-264}$, and Spike$_{577-612}$) were identified in the present study. These sequences will be joined to construct recombinant peptides in the next step. Instead of using inactivated full-length spike protein, we designed a novel recombinant protein construct that increased sequence specificity as well as circumvented mutation sites and glycosylation sites. As the antigen design is based on bioinformatics study, the exact ability of the selected fragments to produce qualified antibodies for virus detection has yet to be determined by experiments.

Noticeably, the spike protein of SARS-CoV-2 is heavily glycosylated. Glycans could shield epitopes during antibody recognition, which may interfere with the detection of viral proteins (*Shajahan et al., 2020*). About 17 N-glycosylation sites along with two O-glycosylation sites were found occupied in the spike protein of SARS-CoV-2 (*Shajahan et al., 2020*). We circumnavigated most glycosylation sites when selecting immunodominant fragments. The three selected fragments in this study only contain 3 glycosylation sites. In case these glycosylation sites impede the diagnostic performance, an additional deglycosylation step with N-glycanase should be applied for the test specimens (*Dermani et al., 2019*), which is a simple and efficient method for deglycosylation (*Hirani, Bernasconi & Rasmussen, 1987*; *Huang et al., 2015*; *Lattová et al., 2016*; *Zheng, Bantog & Bayer, 2011*). Alternatively, an eukaryotic expressing system could be employed to mimic the antigen presented in human cells.

Though coronaviruses can find and repair errors during the replication process (*Wang et al., 2020b*), the SARS-CoV-2 genome still presents a large number of mutations.

Mutations could not only help virus slip past our immune defense, but also spoil the efficiency of diagnostic tests (*Chen et al., 2020b*). In this study, we circumvented high-frequency mutation sites when selecting antigen fragments. In addition, our fragments also avoided RBD regions which are prone to mutation (*Chen et al., 2020b*). The construct finally built contained no high-frequency mutation.

To date, several studies using predictive algorithms to analyze SARS-CoV-2 have been reported (*Alam et al., 2020*; *Behmard et al., 2020*; *Can et al., 2020*; *Chen et al., 2020a*; *Dong et al., 2020*; *Poran et al., 2020*; *Saha, Ghosh & Burra, 2021*; *Sohail et al., 2021*). However, most of these bioinformatics analyses against SARS-CoV-2 intended to develop effective vaccines to prevent infection and the identified sequences possess high homology with other viruses, especially SARS-CoV (*Bhattacharya et al., 2020*; *Chen et al., 2020a*; *Robson, 2020*). On the contrary, the fragments suitable for diagnosis should be unique when compared with other species to ensure the specificity of detection. Therefore, the results obtained from vaccine studies are not ideal for virus detection. In this study, attention was paid to the sequences with high variability, hence the immunodominant fragments identified are more specific. Distinct from vaccine studies, murine MHC alleles were selected in epitopes prediction in this study, so that the designed antigen could trigger a strong humoral immune response in mice. Furthermore, glycosylated sites and recently identified high-frequency mutation sites were deliberately avoided during the screening process to eliminate their potential adverse impact.

In silico analysis has been widely used to mine and identify various pathogens as well as epitopes prediction (*Kiyotani et al., 2020*; *Liò & Goldman, 2004*; *Qin et al., 2003*; *Robson, 2020*; *Shen et al., 2003*). In this study, identified fragments were further scanned in the IEDB database, and found four peptides contained in the sequences were experimentally validated epitopes (Table S5), which reinforced the conclusion of the present study. In the following studies, we will immunize Balb/c mice with the designed antigen to generate mAbs which could be utilized for SARS-CoV-2 diagnosis after evaluating their sensitivity, specificity, and other related properties.

## CONCLUSION

Through bioinformatics analysis, three immunodominant fragments were identified in the present study. After connected by flexible linkers, we acquired a final recombinant peptide with 194 residues. It was predicted to possess high antigenicity and specificity for SARS-CoV-2. Our next move is to express and purify the recombinant protein in a suitable expression system, followed by immunizing the mice with purified immunogen to obtain specific antibodies. The present study would provide aid in developing an antigen-capture based detection system.

### Funding

This work was supported by the emergency and special scientific program of COVID-19 epidemic of Changsha, China (grant number: kq2001038) and the National Science

Foundation of China (grant number: 81970746). The funders had no role in study design, data collection and analysis, decision to publish, or preparation of the manuscript.

## Grant Disclosures
The following grant information was disclosed by the authors:
COVID-19 epidemic of Changsha, China: kq2001038.
National Science Foundation of China: 81970746.

## Competing Interests
The authors declare that they have no competing interests.

## Author Contributions

- Siqi Zhuang conceived and designed the experiments, performed the experiments, analyzed the data, prepared figures and/or tables, authored or reviewed drafts of the paper, and approved the final draft.
- Lingli Tang performed the experiments, authored or reviewed drafts of the paper, and approved the final draft.
- Yufeng Dai performed the experiments, authored or reviewed drafts of the paper, and approved the final draft.
- Xiaojing Feng performed the experiments, authored or reviewed drafts of the paper, and approved the final draft.
- Yiyuan Fang performed the experiments, authored or reviewed drafts of the paper, and approved the final draft.
- Haoneng Tang performed the experiments, authored or reviewed drafts of the paper, and approved the final draft.
- Ping Jiang performed the experiments, authored or reviewed drafts of the paper, and approved the final draft.
- Xiang Wu performed the experiments, authored or reviewed drafts of the paper, and approved the final draft.
- Hezhi Fang performed the experiments, authored or reviewed drafts of the paper, and approved the final draft.
- Hongzhi Chen conceived and designed the experiments, performed the experiments, authored or reviewed drafts of the paper, and approved the final draft.

## Data Availability
   The raw measurements are available in the Supplemental File.

## Supplemental Information
Supplemental information for this article can be found online at http://dx.doi.org/10.7717/peerj.11232#supplemental-information.

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
