# Peer review of "Bioinformatic prediction of immunodominant regions in spike protein for early diagnosis of the severe acute respiratory syndrome coronavirus 2 (SARS-CoV-2)"

_PeerJ, doi:10.7717/peerj.11232_

## Round 0.1 · original submission · Major Revisions

As reviewer 2 indicates, publication of an in silico study of SARS-CoV-2 immune epitopes can be an important first step in assessing identified epitopes in the lab and in vivo. But as reviewer 1 suggested, the novelty of the pipeline presented in this manuscript, and the uniqueness of the results are not readily apparent. Before a decision can be made on the publication of this manuscript in PeerJ, please revise your manuscript taking into account all of the reviewer comments, but in particular, describe the unique aspects of your epitope prediction pipeline that would distinguish your results from previously published work.

Reviewer 1 ·

Basic reporting

The authors performed an epitope level analysis of the spike protein in novel coronavirus or SARS-CoV2. The authors selected these epitopes based on the predictions but did not include the peptides or epitopes that are already confirmed. These epitopes could be easily retrieved from sources like the IEDB.

Experimental design

While the authors concluding the manuscript with the following remark "Our next move is to express and purify the recombinant protein in a suitable expression system, followed by immunizing the mice with purified immunogen to obtain specific antibodies", that shows the interest only in the B cell response. In such a case, prediction to only B cell epitope could have been enough, why did they choose to predict the T cell epitopes too?

Validity of the findings

The authors present preliminary findings or concluded that the validation will be performed in their next move. In the present scenario, the manuscript offers no other validation except for using multiple tools to go with the consensus.

Additional comments

The authors have performed an analysis using a standard set of tools. There are many papers published with pipelines. The authors should consider providing data or discussions on how their analysis/results/interpretation is different from others.

Many of the relevant papers and tools are not referred in the manuscript.

·

Basic reporting

The English language is clear and professional. Also the tables and figures are understandable. The repetations in material/methods and results should be revised.
Minor revisions for introduction;
* Line 45 : Accession date should be added.
* Line 47-48 : Use passive voice in sentence starting with ‘Nevertheless..’
*** Line 84 : ‘The whole flowchart of our work is depicted in Fig. 1.’ It should be added to Material and Methods section.

Experimental design

Experimental design and the selection of bioinformatics tools which made to achieve the aim of the study are sufficient and appropriate to the literature.
Minor revisions for Materials and Method;
* In ‘Data retrieval and sequence alignment’ section, it should be mentioned that alignments were done to find out the SARS-CoV-2 specific genomic regions.
* The accession numbers of sequences (Fig 2) should be added to ‘Data retrieval and sequence alignment’ section.
*The detailed information about servers is not necessary. For example Line 139: VaxiJen was the first server for alignment-independent prediction of protective antigens OR Line 143-145: . ProtScale was one of the protein identification and analysis tools in the ExPASy server, which could calculate hydrophilicity based on amino acid scales.
*The websites of servers and the references should be added.

Validity of the findings

The results of the study were well presented and answered the research question.
Minor revisions for Results;
*Do not repeat the same information which was mentioned in materials and methods. For example Line 177-178 : ‘The antigenicity was calculated via VaxiJen v2.0 with the given cutoff of ≥0.40’
*Line 162: There is no information about performing phylogenetic tree in materials and methods section even a result was mentioned.
*Line 259-261: The result of blast which written as ‘little similarity’ should be mentioned clearly.
*Line 302: ‘an’ instead of ‘a’

Additional comments

This is a well-organized in silico study which needs minor revisions. These preliminary studies are very important for producing early diagnosis approaches in struggling against SARS-CoV-2 pandemic. The authors analysed the coronavirus sequences to find the regions specific to SARS-CoV-2. Then epitopes were predicted in S1 and epitopes out of the glycosylation and highly mutated regions were selected as candidates. Finally, the 3 epitopes were linked with linkers.

---

## Round 0.2 · accepted · Accept

While Reviewer 1 remained critical of this manuscript following revision, I believe that given the stated purpose of the manuscript, the detailed suggestions for further revision do not apply. The goal of your manuscript is to identify epitopes that will elicit an antibody response in mice that will result in antibodies to be used for the development of diagnostics. Therefore, determining human epitopes is not directly pertinent to this goal. And while lab validation would significantly strengthen the paper, it is logical that the current analysis is a prelude to the wet bench experiments that will follow. Therefore, I am happy to accept your manuscript for publication.

Reviewer 1 ·

Basic reporting

The authors have reported a set of epitopes from SARS-CoV2 and suggested a recombinant peptide to elicit the desired immunity.

Experimental design

The experimental design is somewhat questionable. In a computational study, the authors have the choice to work on immune reactivity in humans, and they have chosen a mouse model.
The authors are justifying that choosing a mouse model is one of their novelties. It would be worthwhile to compare the study against human results.

Validity of the findings

The results are computationally predicted and offer no wet lab validation. I appreciate that the authors browsed known epitopes from reference databases like and they found 4 peptides to be matching.

I would suggest authors to expand the search space a little bit and provide a nice comparison between their set of peptides versus epitopes retrieved from the IEDB.

Additional comments

No other comments.